# Machine Learning Consensus Clustering of Hospitalized Patients with Admission Hyponatremia

**DOI:** 10.3390/diseases9030054

**Published:** 2021-08-01

**Authors:** Charat Thongprayoon, Panupong Hansrivijit, Michael A. Mao, Pradeep K. Vaitla, Andrea G. Kattah, Pattharawin Pattharanitima, Saraschandra Vallabhajosyula, Voravech Nissaisorakarn, Tananchai Petnak, Mira T. Keddis, Stephen B. Erickson, John J. Dillon, Vesna D. Garovic, Wisit Cheungpasitporn

**Affiliations:** 1Department of Medicine, Division of Nephrology and Hypertension, Mayo Clinic, Rochester, MN 55905, USA; kattah.andrea@mayo.edu (A.G.K.); Erickson.Stephen@mayo.edu (S.B.E.); Dillon.john@mayo.edu (J.J.D.); garovic.vesna@mayo.edu (V.D.G.); 2Department of Internal Medicine, UPMC Pinnacle, Harrisburg, PA 17105, USA; hansrivijitp@upmc.edu; 3Department of Medicine, Division of Nephrology and Hypertension, Mayo Clinic, Jacksonville, FL 32224, USA; mao.michael@mayo.edu; 4Department of Internal Medicine, Division of Nephrology, University of Mississippi Medical Center, Jackson, MS 39216, USA; pvaitla@umc.edu; 5Department of Internal Medicine, Faculty of Medicine, Thammasat University, Pathum Thani 10120, Thailand; 6Department of Medicine, Section of Cardiovascular Medicine, Wake Forest University School of Medicine, Winston-Salem, NC 27101, USA; saraschandra21@gmail.com; 7MetroWest Medical Center, Department of Internal Medicine, Tufts University School of Medicine, Boston, MA 01760, USA; voravech.niss@gmail.com; 8Division of Pulmonary and Pulmonary Critical Care Medicine, Faculty of Medicine, Ramathibodi Hospital, Mahidol University, Bangkok 10400, Thailand; petnak@yahoo.com; 9Department of Medicine, Division of Nephrology and Hypertension, Mayo Clinic, Phoenix, AZ 85054, USA; keddis.mira@mayo.edu

**Keywords:** artificial intelligence, hyponatremia, sodium, clustering, machine learning, mortality, hospitalization, electrolytes

## Abstract

Background: The objective of this study was to characterize patients with hyponatremia at hospital admission into clusters using an unsupervised machine learning approach, and to evaluate the short- and long-term mortality risk among these distinct clusters. Methods: We performed consensus cluster analysis based on demographic information, principal diagnoses, comorbidities, and laboratory data among 11,099 hospitalized adult hyponatremia patients with an admission serum sodium below 135 mEq/L. The standardized mean difference was utilized to identify each cluster’s key features. We assessed the association of each hyponatremia cluster with hospital and one-year mortality using logistic and Cox proportional hazard analysis, respectively. Results: There were three distinct clusters of hyponatremia patients: 2033 (18%) in cluster 1, 3064 (28%) in cluster 2, and 6002 (54%) in cluster 3. Among these three distinct clusters, clusters 3 patients were the youngest, had lowest comorbidity burden, and highest kidney function. Cluster 1 patients were more likely to be admitted for genitourinary disease, and have diabetes and end-stage kidney disease. Cluster 1 patients had the lowest kidney function, serum bicarbonate, and hemoglobin, but highest serum potassium and prevalence of acute kidney injury. In contrast, cluster 2 patients were the oldest and were more likely to be admitted for respiratory disease, have coronary artery disease, congestive heart failure, stroke, and chronic obstructive pulmonary disease. Cluster 2 patients had lowest serum sodium and serum chloride, but highest serum bicarbonate. Cluster 1 patients had the highest hospital mortality and one-year mortality, followed by cluster 2 and cluster 3, respectively. Conclusion: We identified three clinically distinct phenotypes with differing mortality risks in a heterogeneous cohort of hospitalized hyponatremic patients using an unsupervised machine learning approach.

## 1. Introduction

Sodium is an important cation that contributes to plasma osmolality [1]. In healthy individuals, serum sodium concentration varies by only 1–2% [2]. This process is physiologically regulated by osmoregulation and plasma tonicity regulation [1]. Hyponatremia is the most common electrolyte abnormality in clinical practice associated with significant morbidity, mortality, and healthcare expenditure. Acute hyponatremia can cause malaise, nausea, vomiting, headache, altered mental status, seizures, and ultimately respiratory arrest and coma. Even mild to moderate chronic hyponatremia can have detrimental effects on cognitive function, balance, and insidious bone mineralization loss, with studies showing increased falls, fractures, or even mortality [1,2,3,4,5,6,7,8].

The traditional algorithmic approach to hyponatremia evaluation initially classifies a patient’s volume status as hypovolemic, euvolemic, or hypervolemic [9,10,11]. Investigations to identify the causes of hyponatremia include a thorough history and physical examination followed by laboratory serum and urine studies [3,4]. However, differentiating between euvolemia and hypovolemia can be very clinically challenging, especially on initial hospital admission. In addition, laboratory data including serum osmolarity, urine osmolarity, and urine sodium are usually pending at hospital admission [2,3]. Thus, identifying the phenotype of the hyponatremic patient on hospital admission with its associated impact on evaluation, management, and prognosis can be challenging.

With progress in artificial intelligence, machine learning (ML) has recently been applied in clinical decision support systems [12,13,14]. Consensus clustering is an unsupervised ML approach utilized to identify distinct phenotypes in heterogeneous patient populations. It can be used to assess similarities and differences in large datasets with many variables, and subsequently distinguish patients into novel clusters with distinct phenotypes [12,15,16]. Recent studies have demonstrated that ML consensus clustering can identify disease subtypes that carry different clinical outcomes [17,18].

This study aims to identify distinct phenotypes of patients with hyponatremia on admission from multidimensional data via an unsupervised machine learning approach.

## 2. Materials and Methods

### 2.1. Patient Population

Mayo Clinic Institutional Review Board approved this study. We screened hospitalized patients from 1 January 2011 to 31 December 2013 at Mayo Clinic Hospital, Rochester, Minnesota, USA. If patients had multiple hospital admissions during the study period, we solely analyzed the first admission. We included patients with (1) age ≥18 years and (2) presence of hyponatremia at hospital admission. We defined admission hyponatremia as the first in-hospital serum sodium below 135 mEq/L. We excluded patients with (1) lack of serum sodium measurement within 24 h of hospital admission and (2) no authorization for research use.

### 2.2. Data Collection

We abstracted pertinent demographic information, principal diagnoses, comorbidities, and laboratory data from our hospital’s electronic database, using the previously validated method [2,3,4]. As we aimed to cluster hospitalized hyponatremia patients based on available clinical characteristics at hospital admission, we used only inputted data within 24 h of hospital admission into the analysis. If there were multiple values, we selected the first laboratory value within the 24-h time frame. We excluded variables with over 20% missing data. We imputed missing data for cluster analysis through multiple imputation approach using Random Forest method if variables had missing data of less than 20%.

### 2.3. Clustering Analysis

We applied an unsupervised ML approach to develop clinical phenotypes of hospitalized patients with admission hyponatremia by conducting an unsupervised consensus clustering [19]. We used a pre-specified subsampling parameter of 80% with 100 iterations. The number of potential clusters (k) ranging from 2 to 10, to avoid producing an excessive number of clusters that would not be clinical useful. The optimal number of clusters was determined by examination of the consensus matrix (CM) heat map, cumulative distribution function (CDF), cluster-consensus plots with the within-cluster consensus scores, and the proportion of ambiguously clustered pairs (PAC). The within-cluster consensus score, ranging between 0 and 1, is defined as the average consensus value for all pairs of individuals belonging to the same cluster [20]. A value closer to one indicates better cluster stability. PAC, ranging between 0 and 1, is calculated as the proportion of all sample pairs with consensus values falling within the predetermined boundaries. A value closer to zero indicates better cluster stability. We calculated the PAC using strict criteria with the predetermined boundary of (0, 1), where a pair of individuals who had consensus value greater than 0 or less than 1 was considered ambiguously clustered. Detailed consensus cluster algorithm of this study for reproducibility is provided in the Online Appendix A.

### 2.4. Statistical Analysis

After cluster assignment, we performed analyses to evaluate differences in clinical characteristics and outcomes among the clusters. We compared baseline characteristics among the clusters using an analysis of variance (ANOVA) test for continuous variables and Chi-squared test for categorical variables. To explore the key clinical features of each cluster, we determined the standardized mean differences of clinical characteristics between each cluster and the whole cohort. We regarded variables with absolute standardized mean difference of >0.3 as a key feature of the cluster. Subsequently, we compared hospital mortality and one-year mortality among the clusters. We assessed the association of each cluster membership with hospital mortality using logistic regression and reported odds ratio (OR) with 95% confidence interval (95% CI). We assessed the association of each cluster membership with one-year mortality using Cox proportional hazard regression and reported hazard ratio (HR) with 95% CI. We selected cluster 3 as the reference group for comparison given its lowest mortality risk as well as highest number of patients. We did not adjust for between-group differences in clinical characteristics because these variables were utilized to develop the clusters through unsupervised consensus clustering approach. We performed all analyses using R, version 4.0.3 (RStudio, Inc., Boston, MA, USA. We used the ConsensusClusterPlus package (version 1.46.0) for consensus clustering analysis and the missForest package for missing data imputation.

## 3. Results

Of 76,696 hospitalized adult patients from 2011 to 2013, 11,099 (14%) patients had hyponatremia at hospital admission. Study patients had a mean age of 65 ± 17 years. Approximately half were male, and over 90% were white. The mean admission serum sodium was 131 ± 4 mEq/L and baseline eGFR 73 ± 31 mL/min/1.73 m^2^.

The CDF plot displays the consensus distributions for each k (Figure 1A). The delta area plot shows the relative change in area under the CDF curve (Figure 1B). The largest changes in area occur between k = 3 and k = 4, at which point the relative increase in area becomes noticeably smaller. As shown in the CM heatmap (Figure 1C, Appendix A), the ML algorithm identified cluster 2 and cluster 3 with good cluster stability over repeated iterations.

The mean cluster consensus score was comparable between the scenario of two and three clusters (Figure 2A). Cluster 3 had more favorable low PACs by criteria than cluster 2 (Figure 2B). In summary, using baseline variables at hospital admission, the consensus clustering analysis identified 3 clusters that best represented the data pattern of our hospitalized hyponatremic patients.

Cluster 1 had 2033 (18%) patients, cluster 2 had 3064 (28%) patients, and cluster 3 had 6002 (54%) patients. Table 1 shows the clinical characteristics of the three identified clusters. The distribution of all clinical variables significantly differed among the three clusters.

The standardized mean difference plot was utilized to investigate key features of each cluster, as displayed in Figure 3. According to absolute standardized mean difference of >0.3, cluster 1 patients were more likely to be admitted due to genitourinary disease. Cluster 1 patients also had higher comorbidities (especially diabetes mellitus (DM) and end-stage kidney disease (ESKD)), serum potassium, serum anion gap, and acute kidney injury (AKI), and lower estimated glomerular filtration (eGFR), serum bicarbonate, and hemoglobin. Cluster 2 patients were more likely to be admitted for respiratory disease. Cluster 2 patients were also older, had higher comorbidities (especially coronary artery disease (CAD) congestive heart failure (CHF), stroke, and chronic obstructive pulmonary disease (COPD)), serum bicarbonate, and strong ion difference (SID), but lower serum sodium and serum chloride. Cluster 3 patients were younger, had lower comorbidity burden, and less AKI with corresponding higher eGFR.

Cluster 1 patients had the highest hospital and one-year mortality, followed by cluster 2 and cluster 3 patients (Figure 4A,B), respectively. The OR for hospital mortality was 3.89 (95% CI 2.96–5.11) for cluster 1 and 2.31 (95% CI 1.75–3.05) for cluster 2, when compared to cluster 3. The HR for one-year mortality was 2.35 (95% CI 2.11–2.62) for cluster 1 and 2.01 (95% CI 1.82–2.23) for cluster 2, when compared to cluster 3 (Table 2).

## 4. Discussion

In this study, the unsupervised ML consensus clustering algorithm identified a total of three unique patient clusters based on demographic information, principal diagnoses, comorbidities, and laboratory data. The three clusters represented clinically distinct subgroups derived from multidimensional baseline data.

Cluster 1 consisted of hyponatremic patients with kidney disease (lower baseline eGFR, AKI on hospital admission, and ESKD). These patients had higher serum potassium and anion gap, but lower serum bicarbonate and hemoglobin, which are findings commonly seen in patients with decreased kidney function. Compared to patients in other clusters, cluster 1 patients also had more diabetes and more frequently admitted to the hospital for a principal diagnosis of genitourinary disease. While the degree of hyponatremia in cluster 1 patients was less severe than cluster 2 patients, we found that cluster 1 patients had both the highest in-hospital and one-year mortality among all three clusters. This may represent the prevalent impact of decreased kidney function in this cluster on mortality, which has been well-described in the literature [21,22,23].

The phenotypes of cluster 2 patients included older patients being admitted to the hospital with cardiovascular and respiratory diseases. Among the three clusters, cluster 2 patients had higher CAD, CHF, stroke, and COPD comorbidities. Hyponatremia in the cluster 2 patients was more severe compared to the other clusters. This was in the context of typically a higher serum bicarbonate and SID, normal AGAP, and lower serum chloride. Although cluster 2 patients had lower hospital and one-year mortality compared to cluster 1 patients, they had a two-fold increased risk of mortality compared Cluster 3 patients (who were younger, had lower comorbidity burden, higher eGFR, and the mildest degree of hyponatremia). In the multivariate analysis of 98,411 hospitalized patients, hyponatremia increased the risk in-hospital mortality (OR 1.47; 95% CI 1.33–1.62), one-year mortality (OR 1.38; 1.32–1.46), and 5-year mortality (OR 1.25; 95% CI 1.21–1.30). Unsurprisingly, these relationships were more pronounced in patients admitted with cardiovascular disease and cerebrovascular disease [24].

Our study has several limitations. First, ML consensus cluster analysis utilizes a data-driven approach in which cluster membership relies on the input of the data. This algorithm analyzes shared clinical characteristics from the input variables of examined patients in order to identify potential novel risk factors and phenotypes. Hence, should the imputed variables be changed, cluster membership could be altered. We acknowledge that the majority of our study patient population are Caucasian, and thus this study’s results may not apply to other races or geographic areas. Thus, our study needs validation in other patient populations and in a multicenter study. In addition, our study did not demonstrate the outcome of hyponatremia treatment on attenuation of mortality risk. Further investigations are currently underway. Nevertheless, our study is the first to demonstrate that an unsupervised ML consensus clustering approach using variables that are easy to be obtained on hospital admissions can successfully distinguish meaningful clusters of patients with hyponatremia. In addition, these clusters have different clinical outcomes, and future studies are needed to assess the application of this approach to clinical practice.

## 5. Conclusions

In conclusion, unsupervised ML consensus clustering approach in a heterogenous cohort of hospitalized patients admitted with hyponatremia successfully identified three clinically distinct phenotypes with differing mortality risk. Cluster 1 hyponatremic patients with a suggested phenotype of kidney disease carried the highest risk of hospital and one-year mortality. Future studies are needed to assess whether incorporation of these phenotypes of hyponatremia on admission by the consensus clustering approach can help identify treatment and monitoring strategies to improve patient outcomes.

## Figures and Tables

**Figure 1 diseases-09-00054-f001:**
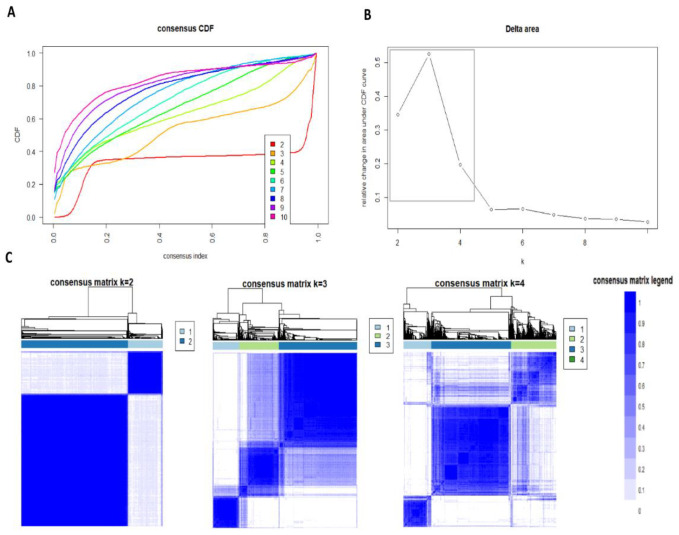
(**A**) CDF plot displaying consensus distributions for each k; (**B**) Delta area plot reflecting the relative changes in the area under the CDF curve. (**C**) Consensus matrix heat map depicting consensus values on a white to blue color scale of each cluster.

**Figure 2 diseases-09-00054-f002:**
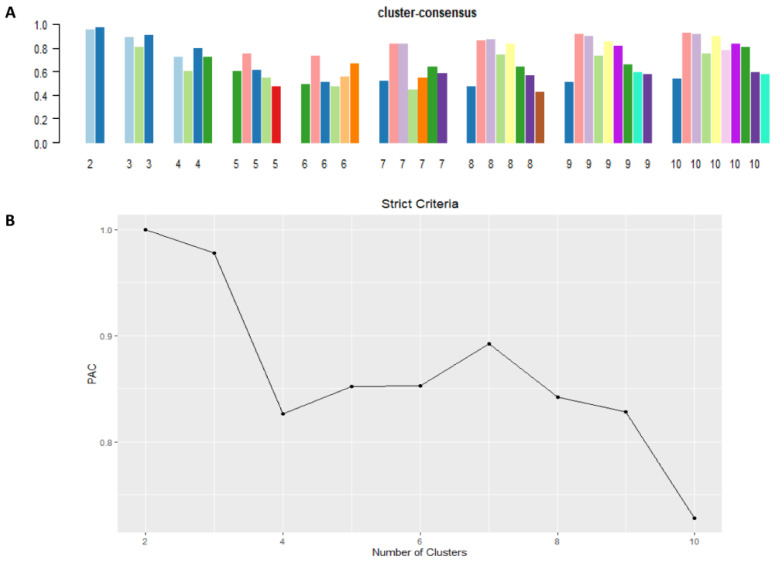
(**A**) The bar plot represents the mean consensus score for different numbers of clusters (K ranges from two to ten); (**B**) The PAC values assessing ambiguously clustered pairs.

**Figure 3 diseases-09-00054-f003:**
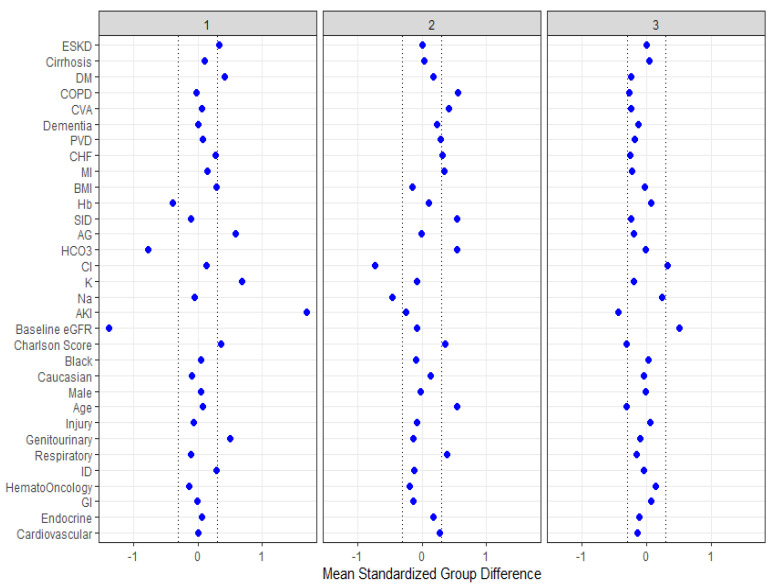
The standardized differences across three clusters for each of baseline parameters. The x axis is the standardized differences value, and the y axis shows baseline parameters. The dashed vertical lines represent the standardized differences cutoffs of <−0.3 or >0.3. Abbreviations: AKI, acute kidney injury; DM, diabetes mellitus; COPD, chronic obstructive pulmonary disease; CVA, cerebrovascular accident; PVD, peripheral vascular disease; CHF, congestive heart failure; MI, myocardial infarction; BMI, body mass index; Hb, hemoglobin; SID, strong ion difference; AG, anion gap; ESKD, end stage kidney disease; HCO3, bicarbonate; Cl, chloride; K, potassium; Na, sodium; GFR, glomerular filtration rate; RS, respiratory system; ID, infectious disease; GI, gastrointestinal.

**Figure 4 diseases-09-00054-f004:**
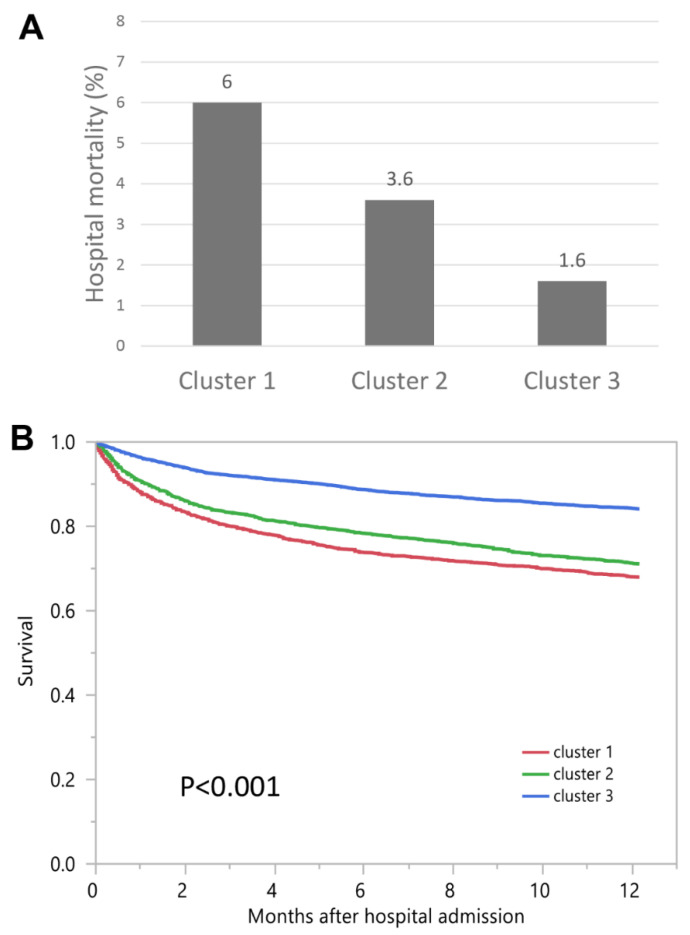
(**A**) Hospital mortality among different clusters of admission hyponatremia; (**B**) One-year mortality among different clusters of admission hyponatremia.

**Table 1 diseases-09-00054-t001:** Clinical characteristics.

Patient Characteristics	Overall	Cluster 1	Cluster 2	Cluster 3	*p*-Value
	(*n* = 11,099)	(*n* = 2033)	(*n* = 3064)	(*n* = 6002)	
Age (years)	65.0 ± 16.8	66.3 ± 15.3	74.2 ± 12.9	59.8 ± 17.0	<0.001
Male sex	5678 (51)	1095 (54)	1538 (50)	3045 (51)	0.02
Race					<0.001
White	10268 (93)	1829 (90)	2943 (96)	5496 (92)	
Black	184 (2)	47 (2)	15 (0.1)	122 (2)	
Others	647 (6)	157 (8)	106 (3)	384 (6)	
BMI (kg/m^2^)	28.5 ± 7.5	30.7 ± 8.7	27.4 ± 6.7	28.3 ± 7.2	<0.001
Principal diagnosis					<0.001
Cardiovascular	1747 (16)	325 (16)	791 (26)	631 (11)	
Endocrine/metabolic	628 (6)	142 (7)	300 (10)	186 (3)	
Gastrointestinal	1382 (12)	247 (12)	237 (8)	898 (15)	
Genitourinary	556 (5)	326 (16)	62 (2)	168 (3)	
Hematology/oncology	1496 (13)	182 (9)	211 (7)	1103 (18)	
Infectious disease	842 (8)	313 (15)	130 (4)	399 (7)	
Respiratory	817 (7)	95 (5)	529 (17)	193 (3)	
Injury/poisoning	1445 (13)	224 (11)	318 (10)	903 (15)	
Other	2186 (20)	179 (9)	486 (16)	1521 (25)	
Charlson Comorbidity Score	2.4 ± 2.7	3.3 ± 2.9	3.3 ± 2.8	1.6 ± 2.2	<0.001
Comorbidities					
Coronary artery disease	959 (9)	257 (13)	560 (18)	142 (2)	<0.001
Congestive heart failure	957 (9)	335 (16)	534 (17)	88 (1)	<0.001
Peripheral vascular disease	454 (4)	116 (6)	300 (10)	38 (0.6)	<0.001
Dementia	198 (2)	38 (2)	150 (5)	10 (0.2)	<0.001
Stroke	971 (9)	213 (10)	626 (20)	132 (2)	<0.001
COPD	1344 (12)	229 (11)	921 (30)	194 (3)	<0.001
Diabetes mellitus	2896 (26)	907 (45)	1036 (34)	953 (16)	<0.001
Cirrhosis	572 (5)	222 (11)	113 (4)	237 (4)	<0.001
End-stage kidney disease	685 (6)	660 (32)	18 (1)	7 (0.1)	<0.001
Laboratory test					
eGFR (mL/min/1.73 m^2^)	73 ± 31	30 ± 18	71 ± 23	89 ± 23	<0.001
Sodium (mEq/L)	131 ± 4	131 ± 3	129 ± 5	132 ± 3	<0.001
Potassium (mEq/L)	4.3 ± 0.7	4.8 ± 0.9	4.3 ± 0.7	4.2 ± 0.6	<0.001
Chloride (mEq/L)	97 ± 5	98 ± 5	93 ± 6	99 ± 4	<0.001
Bicarbonate (mEq/L)	25 ± 4	21 ± 5	27 ± 4	24 ± 3	<0.001
Anion gap	9 ± 4	12 ± 5	9 ± 3	9 ± 3	<0.001
Strong ion difference	38.3 ± 4.0	37.9 ± 4.6	40.4 ± 3.9	37.3 ± 3.3	<0.001
Hemoglobin (g/dL)	11.9 ± 2.2	11.0 ± 2.3	12.1 ± 2.0	12.1 ± 2.2	<0.001
Acute kidney injury	2254 (20)	1793 (88)	308 (10)	153 (3)	<0.001

**Table 2 diseases-09-00054-t002:** Mortality outcomes according to clusters.

	Hospital-Mortality	OR (95% CI)	1-Year Mortality	HR (95% CI)
Cluster 1	6.0%	3.89 (2.96–5.11)	32.0%	2.35 (2.11–2.62)
Cluster 2	3.6%	2.31 (1.75–3.05)	28.9%	2.01 (1.82–2.23)
Cluster 3	1.6%	1 (ref)	15.9%	1 (ref)

## Data Availability

Data is available upon reasonable request to the corresponding author.

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
