# Peer review of "Machine Learning Consensus Clustering of Hospitalized Patients with Admission Hyponatremia"

_diseases, 2021, doi:10.3390/diseases9030054_

Round 1
Reviewer 1 Report
Interesting paper by Thongprayoon and coworkers,
The manuscript is well written, clearly described, the research well planned and executed. The results are sounding and the experimental section well specified.
I recommend accept in present form.
Author Response
Reviewer 1
Interesting paper by Thongprayoon and coworkers,
The manuscript is well written, clearly described, the research well planned and executed. The results are sounding and the experimental section well specified.
I recommend accept in present form.
Response: We thank you for reviewing our manuscript and for your critical evaluation.

Reviewer 2 Report
The purpose of the study was, based on laboratory data from patients with hyponatremia on admission to hospital, to evaluate the short- and long-term prognosis of mortality. More than 11,000 patients with hyponatremia were identified through an unsupervised learning machine approach and divided into related cluster. Three main groups have been identified. Cluster 1: patients with urinary tract infection on admission, diabetes and advanced chronic kidney disease. Cluster 2: Older patients, most with respiratory disease on admission and associated CAD, CHF, stroke and COPD and the lowest level of hyponatremia. Cluster 3: Younger patients, lower degree of comorbidities and better renal function. In a glance evaluation, less informed physicians would bet that the worst prognosis would be cluster 2, related to cardiovascular complications much more frequent and with greater notoriety. The final result showed that cluster 1 had a worse prognosis. The merit of the study was to use an impersonal instrument, with the aid of science (mathematics?) using simple and easily obtainable variables, was possible to make predictions from preliminary data on hospital admission based on the presence of hyponatremia. It should be taken into account that hyponatremia on admission may have been neglected. Patients with hyponatremia and a previous history of cardiovascular complications would be intuitively classified as having a worse prognosis. On the contrary, those with severe and irreversible renal complications had the worst prognosis. Perhaps because being asymptomatic and of the worst knowledge. of doctors in general about kidney disease these complications are less valued. Initiatives such as these, to make clinical prognoses easy to be obtained in hospital admissions through impersonal scientific instruments, can be a warning path with reasonable antecedence so that such patients can be followed up with greater attention in the future.
Author Response
Reviewer 2
The purpose of the study was, based on laboratory data from patients with hyponatremia on admission to hospital, to evaluate the short- and long-term prognosis of mortality. More than 11,000 patients with hyponatremia were identified through an unsupervised learning machine approach and divided into related cluster. Three main groups have been identified. Cluster 1: patients with urinary tract infection on admission, diabetes and advanced chronic kidney disease. Cluster 2: Older patients, most with respiratory disease on admission and associated CAD, CHF, stroke and COPD and the lowest level of hyponatremia. Cluster 3: Younger patients, lower degree of comorbidities and better renal function. In a glance evaluation, less informed physicians would bet that the worst prognosis would be cluster 2, related to cardiovascular complications much more frequent and with greater notoriety. The final result showed that cluster 1 had a worse prognosis. The merit of the study was to use an impersonal instrument, with the aid of science (mathematics?) using simple and easily obtainable variables, was possible to make predictions from preliminary data on hospital admission based on the presence of hyponatremia. It should be taken into account that hyponatremia on admission may have been neglected. Patients with hyponatremia and a previous history of cardiovascular complications would be intuitively classified as having a worse prognosis. On the contrary, those with severe and irreversible renal complications had the worst prognosis. Perhaps because being asymptomatic and of the worst knowledge. of doctors in general about kidney disease these complications are less valued. Initiatives such as these, to make clinical prognoses easy to be obtained in hospital admissions through impersonal scientific instruments, can be a warning path with reasonable antecedence so that such patients can be followed up with greater attention in the future..
Response: We thank you for reviewing our manuscript and for your critical evaluation. We appreciate the reviewer’s kind comments and thus we additionally included important points that the reviewer raised in the discussion of our manuscript as suggested.
“Nevertheless, our study is the first to demonstrate that unsupervised ML consensus clustering approach using variables that are easy to be obtained on hospital admissions can successfully distinguish meaningful clusters of patients with hyponatremia. In addition, these clusters have different clinical outcomes, and future studies are needed to assess the application of this approach to clinical practice.”

Reviewer 3 Report
This is an interesting and well-written study . The authors have managed to distinguish three unique patient clusters , Patient 1cluster with chronic kidney disease these will have worse prognosis , Patient 2 cluster with cardiovascular and respiratory diseases and patient 3 cluster patients of younger age with higher kidney function. This findings will help us categorize the patients and predict mortality and morbidity. I believe the findings are clearly presented and supported. I think it should be accepted at its current status. I congratulate the authors.
Author Response
Reviewer 3
This is an interesting and well-written study. The authors have managed to distinguish three unique patient clusters, Patient 1cluster with chronic kidney disease these will have worse prognosis, Patient 2 cluster with cardiovascular and respiratory diseases and patient 3 cluster patients of younger age with higher kidney function. These findings will help us categorize the patients and predict mortality and morbidity. I believe the findings are clearly presented and supported. I think it should be accepted at its current status. I congratulate the authors..
Response: We thank you for reviewing our manuscript and for your critical evaluation. We greatly appreciated the reviewer’s time and comments.
We greatly appreciated the reviewer’s and editor’s time and comments to improve our manuscript.
